# Ligand-Independent Spontaneous Activation of Purinergic P2Y_6_ Receptor Under Cell Culture Soft Substrate

**DOI:** 10.3390/cells14030216

**Published:** 2025-02-03

**Authors:** Akiyuki Nishimura, Kazuhiro Nishiyama, Tomoya Ito, Xinya Mi, Yuri Kato, Asuka Inoue, Junken Aoki, Motohiro Nishida

**Affiliations:** 1Division of Cardiocirculatory Signaling, National Institute for Physiological Sciences (NIPS), National Institutes of Natural Sciences, Okazaki 444-8787, Japan; 2Exploratory Research Center on Life and Living Systems (ExCELLS), National Institutes of Natural Sciences, Okazaki 444-8787, Japan; 3Department of Physiological Sciences, SOKENDAI (School of Life Science, The Graduate University for Advanced Studies), Okazaki 444-8787, Japan; 4Department of Physiology, Graduate School of Pharmaceutical Sciences, Kyushu University, Fukuoka 812-8582, Japan; knishiyama@omu.ac.jp (K.N.); t.ito@phar.kyushu-u.ac.jp (T.I.); mixinya@phar.kyushu-u.ac.jp (X.M.); yu-kato@phar.kyushu-u.ac.jp (Y.K.); 5Laboratory of Prophylactic Pharmacology, Graduate School of Veterinary Science, Osaka Metropolitan University, Osaka 598-8531, Japan; 6Graduate School of Pharmaceutical Sciences, Tohoku University, Sendai 980-8578, Japan; iaska@tohoku.ac.jp; 7Graduate School of Pharmaceutical Sciences, Kyoto University, Kyoto 606-8501, Japan; 8Department of Health Chemistry, Graduate School of Pharmaceutical Sciences, The University of Tokyo, Tokyo 113-0033, Japan; jaoki@mol.f.u-tokyo.ac.jp

**Keywords:** GPCRs, purinergic receptor, basal activity, Ca^2+^ oscillation, substrate stiffness

## Abstract

G protein-coupled receptors (GPCRs) exist in the conformational equilibrium between inactive state and active state, where the proportion of active state in the absence of a ligand determines the basal activity of GPCRs. Although many GPCRs have different basal activity, it is still unclear whether physiological stresses such as substrate stiffness affect the basal activity of GPCRs. In this study, we identified that purinergic P2Y_6_ receptor (P2Y_6_R) induced spontaneous Ca^2+^ oscillation without a nucleotide ligand when cells were cultured in a silicon chamber. This P2Y_6_R-dependent Ca^2+^ oscillation was absent in cells cultured in glass dishes. Coating substrates, including collagen, laminin, and fibronectin, did not affect the P2Y_6_R spontaneous activity. Mutation of the extracellular Arg-Gly-Asp (RGD) motif of P2Y_6_R inhibited spontaneous activity. Additionally, extracellular Ca^2+^ was required for P2Y_6_R-dependent spontaneous Ca^2+^ oscillation. The GPCR screening assay identified cells expressing 10 GPCRs, including purinergic P2Y_1_R, P2Y_2_R, and P2Y_6_R, that exhibited spontaneous Ca^2+^ oscillation under cell culture soft substrate. Our results suggest that stiffness of the cell adhesion surface modulates spontaneous activities of several GPCRs, including P2Y_6_R, through a ligand-independent mechanism.

## 1. Introduction

G protein-coupled receptors (GPCRs) are the largest family of transmembrane proteins. They are encoded by more than 800 genes in the human genome, and GPCR signaling mediates a wide variety of physiological and pathological events [1]. Therefore, GPCRs are the most commonly addressed drug targets, being the targets of 30–40% of approved drugs [2]. GPCRs precouple to heterotrimeric G protein, and ligand binding leads to conformational changes of GPCRs that catalyze the GDP/GTP exchange on the G protein α (Gα) subunit. Gα-GTP is dissociated from the Gβγ dimer, and Gα-GTP and Gβγ interact with downstream effectors for signal transduction.

Purinergic receptors are a type of receptor that respond to extracellular purines or pyrimidines [3]. Purinergic receptors are divided into two groups: P1 and P2 receptors; the P2 receptors are subdivided into P2X and P2Y subfamilies. Among these categories, P1 and P2Y receptors are GPCRs. P1 receptors are adenosine receptors that are subcategorized into A_1_, A_2A_, A_2B_, and A_3_ receptors. Meanwhile, mammalian P2Y receptors contain eight subtypes (P2Y_1_R, P2Y_2_R, P2Y_4_R, P2Y_6_R, P2Y_11_R, P2Y_12_R, P2Y_13_R, and P2Y_14_R) with different G-protein coupling and ligand selectivity. Among these P2Y receptors, P2Y_6_R couples with the G_q_ and G_12_ family and is mainly activated by uridine diphosphate (UDP). P2Y_6_R plays pivotal roles in physiological and pathological events in cardiovascular fields [4,5]. P2Y_6_R is upregulated in cardiomyocytes by mechanical stress and can be involved in progression of heart failure included by pressure overload [6]. P2Y_6_R contributes to hypertensive vascular remodeling through GPCR heterodimerization with angiotensin II receptor type 1 (AT1R) [7,8]. Experiments using P2Y_6_R-deficient mice also suggest the deleterious role of this receptor in atherosclerosis by promoting inflammation [9,10,11].

GPCRs can be activated by various types of ligands, not only chemical molecules but also physiological stresses such as light, mechanical force, and temperature [12,13]. AT1R was first identified as a mechano-sensitive GPCR directly activated by mechanical force in an angiotensin II-independent manner [14]. Recently, several G_q_-coupled GPCRs such as endothelin ET_A_ receptor, muscarinic M_5_ receptor, vasopressin V_1A_ receptor, histamine H1 receptor, bradykinin B2 receptor, sphingosine 1-phosphate receptor, dopamine D5 receptor, and GPR68 receptor have been reported as mechano-sensitive GPCRs [15,16,17,18,19,20]. These mechano-sensitive GPCRs have critical roles in vascular function. Shear stress is the tangential force from blood flow on the endothelial surface of the blood vessel, and several mechano-sensitive GPCRs sense shear stress to control vasodilation [21]. Mechano-sensitive GPCRs are also involved in hemodynamic load-induced cardiac hypertrophy [14] and preeclampsia [22].

Because P2Y_6_R forms a heterodimer with mechano-sensitive AT1R [7,8] and mediates mechanical stress-induced fibrogenic factor formation in cardiomyocytes [6], we speculated mechano-sensitivity of P2Y_6_R. However, when P2Y_6_R-expressing HeLa cells were seeded onto a silicon/polydimethylsiloxane (PDMS)-based stretch chamber, spontaneous Ca^2+^ oscillation was observed without mechanical stimulation. In this study, we investigated the characterization of GPCR-mediated Ca^2+^ oscillation under cell culture soft substrates.

## 2. Materials and Methods

### 2.1. Plasmid Construction

The mouse P2Y receptor (P2Y_1_R, P2Y_2_R, P2Y_4_R, P2Y_6_R, P2Y_12_R, P2Y_13_R, and P2Y_14_R) genes were amplified by PCR and cloned in pcDNA3 vector. P2Y_6_R mutants were generated by site-directed PCR mutagenesis. For inactive P2Y_6_R AAY mutants, Gln^123^ and Arg^124^ were replaced with Ala. The extracellular RGD motif of P2Y_6_R was mutated with RGE by replacing Asp^90^ with Glu. For GPCR library plasmids, full-length human GPCR genes were cloned into the pCAGGS or pcDNA3.1 vectors [23].

### 2.2. Cell Culture and Transfection

HeLa (ATCC Cat# CCL-2) cells were cultured in DMEM (high glucose) supplemented with 10% fetal bovine serum (FBS) and 1% penicillin/streptomycin (P/S). Plasmid DNA was transfected using Viafect (Promega, Madison, WI, USA). pcDNA3 vector was used as a control plasmid.

### 2.3. Intracellular Ca^2+^ Imaging Using Fura-2

HeLa cells were used for Ca^2+^ oscillation analysis, because treatment of histamine induces prolonged Ca^2+^ oscillation in HeLa cells; its mechanism includes the involvement of IP_3_ receptor and Ca^2+^-induced Ca^2+^ release (CICR) [24,25]. The silicon chambers (stretch chamber, 4 cm^2^, Menicon Life Science, Aichi, Japan) and glass-bottom dishes were coated with collagen (Cellmatrix Type I-C, Nitta Gelatin, Osaka, Japan), laminin (Fujifilm Medical, Tokyo, Japan), or fibronectin (Fujifilm Medical) for 3 h at 37 °C. HeLa cells transfected with P2Y_6_R or its mutant were trypsinized and seeded onto a substrate-coated silicon chamber or glass-bottom dish. The next day, cells were loaded with 5 µM Fura-2 AM (Dojindo, Kumamoto, Japan) for 30 min at 37 °C in DMEM. The dye solution was then replaced with HEPES-buffered saline solution (HBSS) containing 10 mM HEPES pH 7.4, 118 mM NaCl, 5.4 mM KCl, 1.2 mM MgCl_2_, 2 mM CaCl_2_, and 10 mM glucose). The silicon chamber or dish was mounted on the stage of an upright fluorescence microscope (Olympus, Tokyo, Japan). Fura-2 signals were recorded and analyzed using a video image analysis system (Aquacosmos, Hamamatsu Photonics, Shizuoka, Japan) (Figure 1A). P2Y_6_R-specific agonist 10 µM 3-pUDP was added for the indicated timing. The 3-pUDP-responsible cells were used for Ca^2+^ oscillation analysis as P2Y_6_R-positive cells, and the population of Ca^2+^ oscillating cells among the P2Y_6_R-positive cells was calculated. For the P2Y_6_R-specific antagonist MRS2578 experiment, cells co-transfected with P2Y_6_R and GFP were incubated with 10 µM MRS2578 in HBSS for 30 min before imaging. Fura-2 signals of GFP-positive cells were recorded. For P2Y_6_R AAY and other P2Y receptor experiments, HeLa cells co-transfected with each P2Y receptor and GFP were cultured in a collagen-coated silicon chamber. To reduce the proportion of cells expressing only GFP, a 10-fold higher amount of receptor plasmid was transfected compared with GFP plasmid. Fura-2 signals of GFP-positive cells were recorded, and the population of Ca^2+^ oscillating cells among the GFP-positive cells was calculated. About 50 successfully transfected cells in the field of view of the microscope were analyzed per experiment.

### 2.4. Intracellular Ca^2+^ Imaging Using GCaMP6

GCaMP is a calmodulin-based genetically encoded fluorescent calcium indicator, and GCaMP6 shows superior brightness with high calcium sensitivity, detecting rapid Ca^2+^ transients and oscillation [26]. HeLa cells co-transfected with the GPCR library and GCaMP6 were seeded onto a collagen-coated silicon chamber. To reduce cells expressing only GCaMP6, a 10-fold higher amount of GPCR library plasmid was transfected compared with the GCaMP6 plasmid. The next day, the medium was replaced with HBSS, and the silicon chamber was mounted on the stage of an upright fluorescence microscope. GCaMP fluorescence intensity was measured using the GFP filter set and analyzed.

### 2.5. Western Blotting

Cells were lysed with a lysis buffer (20 mM HEPES pH 7.4, 100 mM NaCl, 1 mM EDTA, 1% Triton X-100, and 0.1% SDS) with protease inhibitor cocktail. The lysate was centrifuged (16,000× *g* for 15 min at 4 °C), and an aliquot of the supernatant (10 µg of protein) was mixed with 2× Laemmli buffer with DTT. Samples were subjected to SDS–polyacrylamide gel electrophoresis and transferred onto PVDF membrane. The membrane was blocked in 2% BSA in Tris-buffered saline with Tween-20 (TBS-T) and then incubated with anti-G_q_α (sc-136181, Santa Cruz, Dallas, TX, USA) or anti-GAPDH (016-25523, Fujifilm Medical) antibodies overnight at 4 °C, followed by HRP-conjugated mouse IgG (7076, Cell Signaling Technology, Danvers, MA, USA). Blots were developed with Clarity Max Western ECL Substrate (Bio-rad, Hercules, CA, USA) and detected using ImageQuant LAS4000 Ver1.2 (Cytiva, Marlborough, MA, USA). Protein band intensity was measured using ImageQuant TL software Ver8.1.

### 2.6. Statistical Analysis

We performed statistical analysis using GraphPad Prism 9.0 (GraphPad Software, LaJolla, CA, USA). Results are presented as mean ± SEM from at least 3 independent experiments. Statistical comparisons were determined using a two-tailed Student’s *t*-test (for two groups) or one-way ANOVA with Tukey’s post hoc test (for three or more groups).

## 3. Results

### 3.1. P2Y6R-Mediated Spontaneous Ca^2+^ Oscillation Under Silicon Chamber Culture Conditions

To analyze P2Y_6_R activity, HeLa cells were transiently transfected with P2Y_6_R-expressing plasmid, and intracellular Ca^2+^ dynamics were then measured using a ratiometric fluorescence dye, Fura-2 AM. When P2Y_6_R-expressing HeLa cells were cultured in a collagen-coated silicon/PDMS chamber, Ca^2+^ oscillation at a constant rhythm was observed without nucleotide ligand administration (Figure 1B, Appendix A). Treatment with the P2Y_6_R-specific agonist 3-phenacyl-UDP (3-pUDP) rapidly and transiently increased intracellular Ca^2+^ in oscillating and non-oscillating cells (Figure 1B). When P2Y_6_R-expressing HeLa cells were cultured in a collagen-coated glass bottom dish, spontaneous Ca^2+^ oscillation was not observed, whereas 3-pUDP treatment induced a transient Ca^2+^ increase (Figure 1C). Moreover, when HeLa cells transfecting the control plasmid were cultured in a collagen-coated silicon chamber, Ca^2+^ oscillation was not observed (Figure 1D). These results suggest that P2Y_6_R is spontaneously activated without agonist administration when cells are cultured onto a cell culture soft substrate such as PDMS. About 30% of cells showed Ca^2+^ oscillation at various amplitudes (Figure 1B,E).

### 3.2. Effet of Extracellular Matrix Proteins on P2Y_6_R-Mediated Ca^2+^ Oscillation

The extracellular matrix is important for mechanotransduction and cellular responses [27,28]. Collagens are the most abundant feature of the extracellular matrix, whereas fibronectin and laminin, which are noncollagenous glycoproteins, are also known as major components of the basal membrane. To investigate whether different types of extracellular matrix proteins would affect spontaneous Ca^2+^ oscillation, we tested not only collagen (Figure 1) but also laminin and fibronectin. P2Y_6_R-expressing HeLa cells were cultured in a laminin or fibronectin-coated silicon chamber. The population of oscillating cells was almost 20–30% in both the laminin or fibronectin-coated silicon chamber (Figure 2A,B), which was almost the same as with the collagen coating (Figure 1E). These results suggest that type of substrate does not affect P2Y_6_R-mediated spontaneous Ca^2+^ oscillation.

### 3.3. G Protein-Mediated Signaling and Ca^2+^ Entry from Extracellular Space Is Required for Spontaneous Ca^2+^ Oscillation of P2Y_6_R

Intracellular Ca^2+^ increase is mainly triggered by Ca^2+^ release from the ER or extracellular space. To investigate whether Ca^2+^ entry from the extracellular space through Ca^2+^ channels is required for spontaneous Ca^2+^ oscillation, Ca^2+^ imaging was performed in HBSS buffer without Ca^2+^. The 3-pUDP treatment still increased the intracellular Ca^2+^ because P2Y_6_R-mediated G_q_α activation induced Ca^2+^ release from the ER through inositol 1,4,5-triphosphate (IP_3_) receptors (Figure 3A). However, P2Y_6_R-mediated Ca^2+^ oscillation under a collagen-coated silicon chamber completely disappeared using HBSS without Ca^2+^ (Figure 3A), suggesting that Ca^2+^ entry from the extracellular space is required for spontaneous Ca^2+^ oscillation.

G protein is the main signal transducer of GPCR signaling pathways. The DRY motif of GPCR is important for coupling to G proteins and receptor activation [29]. The AAY mutant of AT1R does not couple to G protein but still can transduce β-arrestin signaling [30]. HeLa cells expressing the P2Y_6_R AAY mutant were cultured in a collagen-coated silicon chamber. P2Y_6_R AAY mutants lost spontaneous Ca^2+^ oscillation (Figure 3B). As a control experiment, Ca^2+^ transients induced by 3-pUDP disappeared and ATP-mediated Ca^2+^ transients through other P2Y receptors were observed (Figure 3B). This result indicates that G protein-mediated signaling is required for spontaneous Ca^2+^ oscillation. The amount of G protein is important for the intensity of downstream signaling. We checked G_q_α expression levels and found that G_q_α expression was not increased in Ca^2+^ oscillating cells (Figure 3C).

Locally released nucleotides activate P2Y receptors in an autocrine/paracrine manner [31]. To investigate the possibility that cells cultured in a collagen-coated silicon chamber would spontaneously release UDP for P2Y_6_R activation, we ran the antagonist experiment. MRS2578 is a selective antagonist of P2Y_6_R and blocks UDP-induced Ca^2+^ transients [32]. To test the effect of MRS2578, cells were treated with MRS2578 for 30 min before Ca^2+^ imaging. P2Y_6_R-mediated Ca^2+^ oscillation was observed under MRS2578 treatment (Figure 3D). Similar to untreated conditions, about 30% of cells expressing P2Y_6_R still have spontaneous Ca^2+^ oscillation under MRS2578 treatment (Figure 3E). This result would exclude the possibility that P2Y_6_R was activated by spontaneously released nucleotides from cells.

### 3.4. The Extracellular RGD Motif of P2Y_6_R Is Required for Spontaneous Ca^2+^ Oscillation

The RGD motif is a cell adhesion sequence that binds to integrin [33]. P2Y_6_R has the RGD motif in the first extracellular loop (Figure 4A). The mutation of the RGD sequence to RGE causes the loss of the ability to bind to integrin [34]. P2Y_2_R associates with α_v_β3/β5 integrin and P2Y2R RGE mutant reduces its binding [35]. P2Y_6_R RGE-expressing cells under a collagen-coated silicon chamber showed Ca^2+^ transients after 3-pUDP administration (Figure 4B) and their responsiveness was almost the same as that of P2Y_6_R WT (Figure 4C), indicating that the P2Y_6_R RGE mutant is functional. On the other hand, spontaneous Ca^2+^ oscillation was not observed in the P2Y_6_R RGE mutant, suggesting the critical role of the RGD motif of P2Y_6_R in relation to Ca^2+^ oscillation (Figure 4B).

### 3.5. Ligand-Independent Ca^2+^ Oscillation Is Observed in Several GPCRs

There are eight mammalian P2Y receptors; among these, P2Y_2_R has the RGD motif in the extracellular loop region, like P2Y_6_R. To investigate whether spontaneous Ca^2+^ oscillation would be observed in other P2Y receptors, HeLa cells were transfected with a plasmid expressing each P2Y receptor and cultured in a collagen-coated silicon chamber. The RGD motif containing P2Y_2_R induced spontaneous Ca^2+^ oscillation, albeit in a slightly irregular rhythm (Figure 5A). Moreover, P2Y_1_R that did not have the RGD motif also showed obvious Ca^2+^ oscillation (Figure 5A). About 10% of cells expressing P2Y_1_R showed spontaneous Ca^2+^ oscillation (Figure 5B). On the other hand, other P2Y receptors including P2Y_4_R, P2Y_12_R, P2Y_13_R, and P2Y_14_R did not induce Ca^2+^ oscillation.

To verify how many GPCRs showed spontaneous Ca^2+^ oscillation, HeLa cells were transfected with GPCR plasmid library and Ca^2+^ indicator GCaMP6 and cultured in a collagen-coated silicon chamber. Using 259 GPCR-expressing plasmids, we monitored GCaMP6 intensity for 5 min and oscillation frequency was counted. Among 259 GPCRs, spontaneous Ca^2+^ oscillation was observed in 10 GPCRs, including P2Y_6_R, P2Y_1_R, P2Y_2_R, prostaglandin E2 receptor 1 (EP1), growth hormone secretagogue receptor (GHSR), histamine H1 receptor (H1R), platelet-activating factor receptor (PAFR), Ca^2+^-sensing receptor (CaSR), gastrin-releasing peptide receptor (GRPR), and Mas-related GPCR, member H (MRGH) (Figure 5C). It was observed that oscillation patterns were different for each GPCR (Figure 5D).

## 4. Discussion

In this study, we investigated the nucleotide ligand-independent activity of P2Y_6_R and found that P2Y_6_R induced spontaneous Ca^2+^ oscillation when cells were cultured in a silicon chamber (Figure 1). Nucleotides can be released from cells by two different pathways: exocytotic release from secretory pathways and conductive/transport pathways [36]. Some central and peripheral neurons release ATP via neuronal synaptic vesicles via purinergic neurotransmission. Multiple channels including connexin (Cx) hemichannels, pannexin 1 (PANX1), volume-regulated anion channels (VRACs), calcium homeostasis modulator 1 (CALHM1), and maxi-anion channels (MACs) have been reported to regulate ATP release [37]. Some Cx hemichannels [38], PANX1 [6], and CALHM1 [39] mediate mechanical stress-induced ATP release. MACs mediate hypoosmotic stress-induced ATP release in astrocytes [40]. In addition, we previously found that mechanical stress induced ATP and UDP release from cardiomyocytes through PANX1, and released nucleotides activated P2Y_6_R to induce fibrotic responses in an autocrine or paracrine manner [6]. This evidence leads to speculation on the possibility that locally released nucleotides continuously activate P2Y_6_R for Ca^2+^ oscillation in HeLa cells cultured in a silicon chamber, although nucleotide release in response to substrate stiffness has not been reported. The P2Y_6_R antagonist MRS2578, which inhibits UDP-induced Ca^2+^ increases, did not prevent Ca^2+^ oscillation under a silicon chamber (Figure 3D). This result would exclude the possibility that P2Y_6_R is continuously activated by released nucleotides from cells in response to substrate stiffness.

The GPCR screening assay revealed that spontaneous activity under a silicon chamber was observed in 10 GPCRs of 259 GPCRs we examined (Figure 5). Because we used the screening system to detect intracellular Ca^2+^ dynamics, all hit GPCRs were G_q_-coupled receptors. cAMP oscillation has been observed in several biological events [41], and extracellular matrix such as laminin can regulate cAMP signaling of G_s_-coupled GPCRs [42,43]. Novel screening systems detecting other second messengers such as cAMP or individual G-protein activity using fluorescence resonance energy transfer (FRET) or bioluminescence resonance energy transfer (BRET)-based biosensors [44,45] may additionally identify GPCRs activated under a silicon chamber. Substrate stiffness is one of the important physical factors that determine cellular response and function [46,47]. PDMS has frequently been used to analyze cellular response in relation to substrate stiffness. P2Y_6_R and others may respond to substrate stiffness, especially soft substrate. It has been reported that mechano-sensors such as integrins, Piezo1, Notch, and TRPV4 recognize substrate stiffness [48,49,50]. GPCRs activated by mechanical force have been well investigated, and among the 10 hit GPCRs, the H1 receptor is known as a mechanosensitive GPCR [16]. However, little is known about the role of GPCRs in cellular response to substrate stiffness. Isoproterenol-induced β2 adrenergic receptor-PKA signaling decreased under cell culture soft substrate [51]. In the current study, we observed that the basal activity of GPCRs in the absence of ligands was affected by substrate stiffness. As far as we know, the relationship between the basal activity of GPCR and substrate stiffness has not been mentioned before.

Recent biochemical and structural studies of GPCRs reveal the conformational dynamics of GPCRs that contribute to their activation. In the absence of ligands, GPCRs are considered to exist in dynamic equilibrium between the inactive and active conformation, and agonist binding shifts the equilibrium toward the active conformation [52]. The proportion of the active conformation that exists in the ligand-absent condition determines the basal activity of GPCRs [53,54]. Many GPCRs show varying degrees of basal activity. Binding with agonists or inverse agonists, receptor dimerization, and some mutations can affect the basal activity of GPCRs [55,56]. Some adhesion GPCRs can associate with extracellular matrix components and regulate their activity [57]. It is quite reasonable that the mutation of the RGD motif of P2Y_6_R led to the loss of spontaneous Ca^2+^ oscillation, because the RGD motif is important for the interaction with integrin [34] which is a well-known stiffness sensor [48]. However, the fact that only P2Y_6_R and P2Y_2_R from among the 10 GPCRs share the RGD motif suggests that the molecular mechanism of spontaneous Ca^2+^ oscillation under a silicon chamber is more complex. We compared the amino acid sequences of 10 GPCRs, but there was no obvious consensus feature among the amino acid sequence alignments. More detailed structural and functional analysis will be required for the characterization of these 10 GPCRs.

Ca^2+^ is the most fundamental regulator of various cellular responses. Signaling patterns of Ca^2+^ include spike, wave, and oscillation. Oscillation is the most complex of these, and Ca^2+^ entry and release are continuously repeated within a suitable balance [58,59]. Ca^2+^ enters from the ER and extracellular space. All 10 GPCRs that we identified in this study are G_q_-coupled GPCRs. G_q_ signaling induces Ca^2+^ release from the ER through IP_3_ receptors. Activation of G_q_α also induces receptor-operated Ca^2+^ entry from extracellular space through Ca^2+^ channels such as transient receptor potential canonical (TRPC) [60]. Treatment of G_q_-coupled GPCR agonists such as histamine and UTP induced ligand-triggered prolonged Ca^2+^ oscillation in HeLa cells [61,62], its mechanism in HeLa cells has been discussed to some extent. Ca^2+^ oscillation is initially dependent on the IP_3_ receptor and not external Ca^2+^, whereas transmembrane Ca^2+^ flux is critical for Ca^2+^ oscillation during the latter phase [24,25]. Because external Ca^2+^ and G_q_ signaling were required for spontaneous Ca^2+^ oscillation under a silicon chamber (Figure 3A,B), it is expected that there are many similarities in the mechanisms of ligand-triggered and ligand-independent Ca^2+^ oscillations.

In this study, we first found that several GPCRs, including P2Y_6_R, spontaneously activated under cell culture soft substrate. Representing a limitation of this study, the molecular mechanism of spontaneous Ca^2+^ oscillation of GPCRs under cell culture soft substrate is still mostly unclear, and its elucidation is a major issue for the future. About 30% of P2Y_6_R-expressing HeLa cells showed spontaneous Ca^2+^ oscillation (Figure 1D). Because HeLa cells are from heterogenous cellular origin and have a variable genome and transcriptome at single-cell level [63], only some HeLa cell populations have complete components for spontaneous Ca^2+^ oscillation. Comparison of gene expression patterns between oscillated and non-oscillated HeLa cell clones would help to identify the molecular mechanism of spontaneous Ca^2+^ oscillation. Because the frequency of Ca^2+^ oscillation depends on cell type [64], validation of spontaneous activation of P2Y_6_R within various cell lines would also contribute to identification of the mechanism. Additionally, the surrounding environment of oscillating cells, including cell–cell interaction, may be involved in spontaneous activity, because the P2Y_6_R RGD motif integrin-binding site was required for Ca^2+^ oscillation (Figure 4). Detailed analysis of the frequency and pattern of Ca^2+^ oscillation at different cell densities would also contribute to identifying the molecular mechanism.

Confirmation of spontaneous Ca^2+^ oscillation of endogenous P2Y_6_R and other examples in primary cells is important for proving the physiological and pathophysiological meaning of GPCR spontaneous activity. The extracellular matrix has important roles in various aspects of cardiovascular physiology, such as heart development, and pathophysiology, such as heart failure progression [65,66]. Cardiac fibrosis, which refers to excess deposition of the extracellular matrix, is considered a common feature in various types of heart failure, and increased stiffness associated with cardiac fibrosis contributes to cardiac dysfunction and heart failure progression. P2Y_6_R expression is upregulated in the pressure-overloaded heart, which contributes to cardiac fibrosis formation [6]. Additionally, cardiomyocyte-specific overexpression of P2Y_6_R exacerbates pressure overload-induced heart failure [67]. In addition to P2Y_6_R, the mRNA of P2Y_2_R is upregulated in congestive heart failure and P2Y_2_R-mediated signaling contributes to cardiac fibrosis [68]. P2Y_1_R is involved in shear stress-mediated mechanotransduction in atrial myocytes and endothelial cells [69,70]. Additionally, it has been reported that P2Y_6_R promotes cancer progression and metastasis [71]. Since matrix stiffness is critical for the progression of various types of cancer [72], the spontaneous activity of GPCRs may contribute to cancer formation. To investigate the physiological and pathophysiological meaning of spontaneous GPCR activation under a silicon chamber, it will be important to analyze whether spontaneous GPCR activation is observed in more physiological conditions such as tissue slices and 3D hydrogel-based organ-like culture.

## 5. Conclusions

In this study, we found that some GPCRs, including purinergic P2Y_6_R, showed spontaneous Ca^2+^ oscillation without ligand stimulation when cells were cultured in a silicon chamber. However, its molecular mechanism is still largely unknown. Because various GPCRs are directly activated by mechanical forces such as shear stress, the similarities and differences in the their stiffness and shear stress sensing mechanisms, and the relationships between these, are important issues. Basal activity of GPCRs is important for various biological events, and disease-causing mutations of GPCRs with increased basal activity have been reported [54]. Future analysis to identify the molecular mechanism of spontaneous GPCR activity under cell culture soft substrate would provide novel insight into GPCR-mediated stiffness regulation and its physiological role.

## Figures and Tables

**Figure 1 cells-14-00216-f001:**
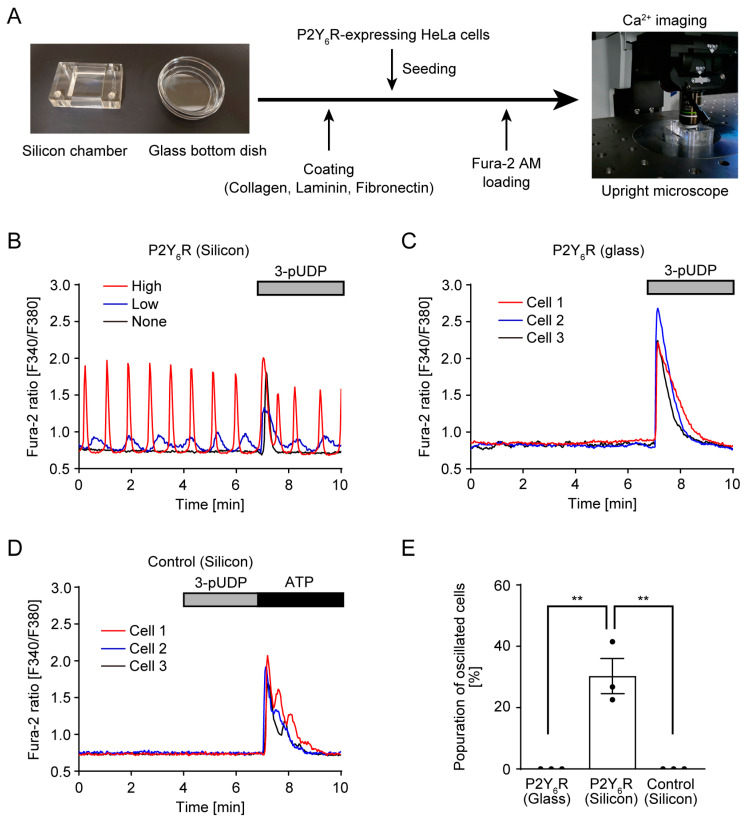
Spontaneous Ca^2+^ oscillation by P2Y_6_R-expressing HeLa cells cultured in a collagen-coated silicon chamber: (**A**) Schematic of Ca^2+^ imaging analysis of cells cultured in substrate-coated silicon or glass-bottom dish; (**B**) Trace of Ca^2+^ response in P2Y_6_R-expressing HeLa cells cultured in a collagen-coated silicon chamber. P2Y_6_R-expressing HeLa cells were seeded on a collagen-coated silicon chamber and the intracellular Ca^2+^ response was monitored via a Fura-2 probe. Representative traces of oscillating cells with high (red) and low (blue) amplitude and non-oscillating cells (black) are shown. P2Y_6_R-specific agonist 10 µM 3-phenacyl UDP (3-pUDP) was added at the indicated time; (**C**) Trace of Ca^2+^ response in P2Y_6_R-expressing HeLa cells cultured in a collagen-coated glass-bottom dish; 10 µM 3-pUDP was added at the indicated time; (**D**) Trace of Ca^2+^ response in control HeLa cells cultured in a collagen-coated silicon chamber; 10 µM 3-pUDP and 100 µM ATP were added at the indicated time; (**E**) Quantitative population of oscillating cells (n = 3 independent experiments). Data are shown as means ± SEM. ** *p* < 0.01, one-way ANOVA.

**Figure 2 cells-14-00216-f002:**
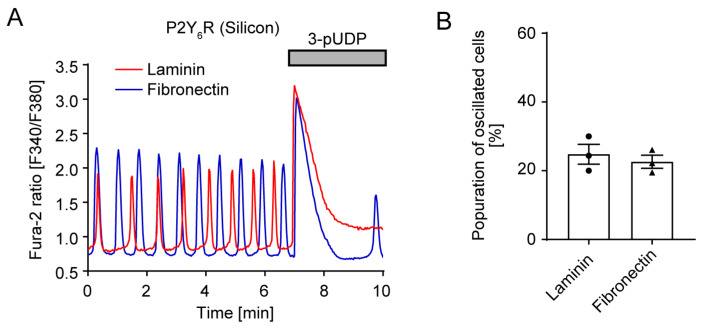
Effects of chamber coating on P2Y_6_R-mediated Ca^2+^ oscillation: (**A**) Trace of Ca^2+^ response in P2Y_6_R-expressing HeLa cells cultured in a laminin or fibronectin-coated silicon chamber. P2Y_6_R-expressing HeLa cells were seeded on a laminin (red) or fibronectin (blue)-coated silicon chamber, and 10 µM 3-pUDP was added at the indicated time. (**B**) Quantitative population of oscillating cells (n = 3 independent experiments). Data are shown as means ± SEM.

**Figure 3 cells-14-00216-f003:**
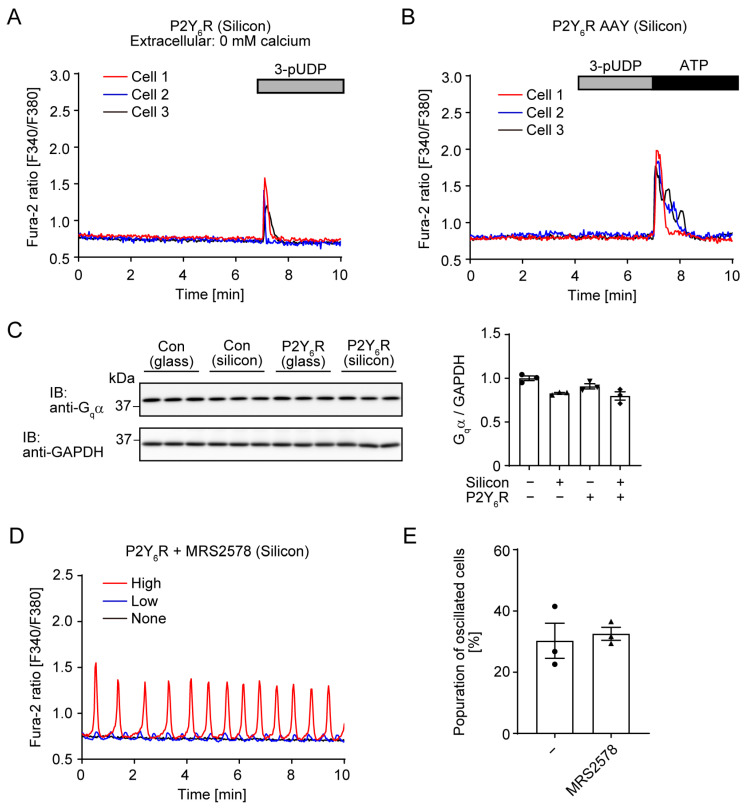
Conditions of P2Y_6_R-mediated spontaneous Ca^2+^ oscillation: (**A**) Effect of extracellular Ca^2+^ on P2Y_6_R-mediated Ca^2+^ oscillation. P2Y_6_R-expressing HeLa cells were cultured in a collagen-coated silicon chamber and Fura-2 imaging was performed in HBSS without Ca^2+^; 10 µM 3-pUDP was added at the indicated time (n = 3 independent experiments); (**B**) Trace of Ca^2+^ response in P2Y_6_R AAY-expressing HeLa cells cultured in a collagen-coated silicon chamber; 10 µM 3-pUDP and 100 µM ATP were added at the indicated time (n = 3 independent experiments); (**C**) Expression of G_q_α in P2Y_6_R or control vector-expressing HeLa cells cultured in a collagen-coated silicon chamber or glass-bottom dish. Molecular weights are shown. Right graph is the quantification of G_q_α expression normalized by GAPDH expression (n = 3 independent experiments); (**D**) Effect of MRS2578 on P2Y_6_R-mediated Ca^2+^ oscillation. P2Y_6_R-expressing HeLa cells were cultured in a collagen-coated silicon chamber, and 10 µM MRS2578 was added 30 min before Ca^2+^ imaging (**E**) Quantitative population of oscillating cells. Data for MRS2578 (−) are from Figure 1E (n = 3 independent experiments). Data are shown as means ± SEM.

**Figure 4 cells-14-00216-f004:**
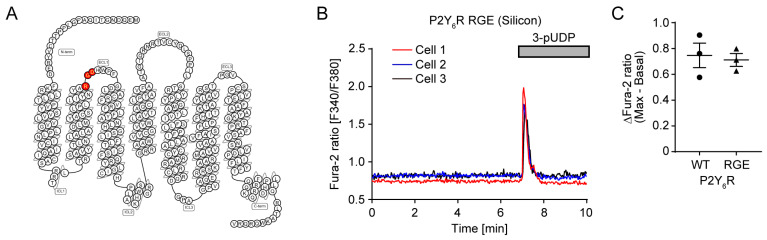
Effect of the extracellular RGD motif on P2Y_6_R-mediated Ca^2+^ oscillation: (**A**) Snake diagram of the mouse P2Y_6_R. The RGD motif is highlighted in red. This diagram was prepared using the GPCR database (http://www.gpcrdb.org, accessed on 19 December 2024); (**B**) Trace of Ca^2+^ response in P2Y_6_R RGE-expressing HeLa cells cultured in a collagen-coated silicon chamber; 10 µM 3-pUDP was added at the indicated time (n = 3 independent experiments); (**C**) Summary of maximum change in Fura-2 ratio after 3-pUDP treatment in HeLa cells expressing P2Y6R WT or RGE mutant. Data for P2Y6R WT are from Figure 1B (n = 3 independent experiments).

**Figure 5 cells-14-00216-f005:**
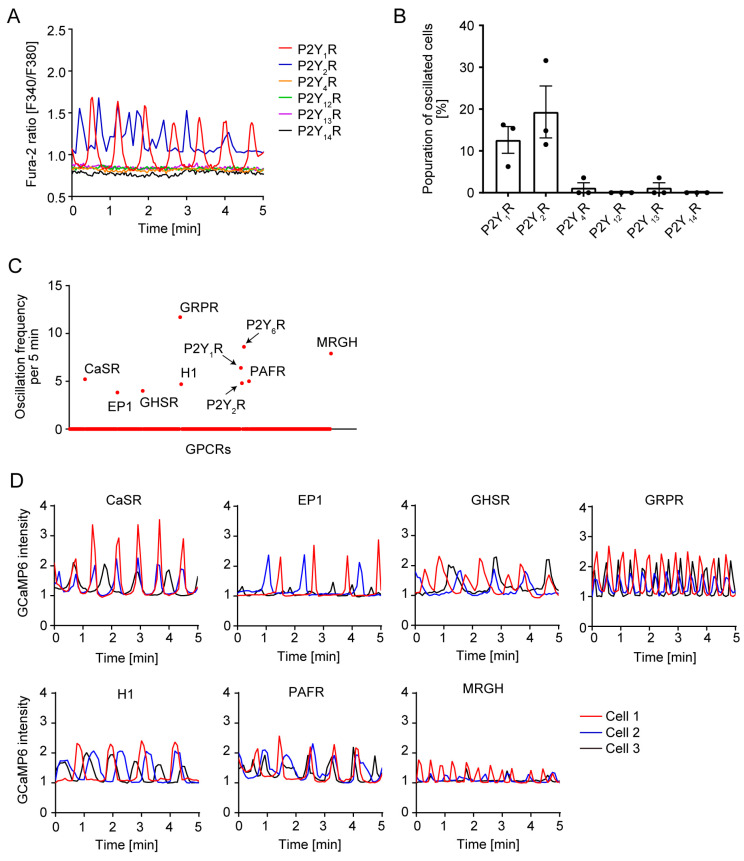
Screening of GPCRs with spontaneous Ca^2+^ oscillation under a collagen-coated silicon chamber: (**A**) Trace of Ca^2+^ response in HeLa cells expressing each P2Y receptor cultured in a collagen-coated silicon chamber; (**B**) Quantitative population of oscillating cells (n = 3 independent experiments); (**C**) Screening of GPCRs showing Ca^2+^ oscillation. HeLa cells expressing GPCR library (259 GPCRs) and Ca^2+^ indicator GCaMP6 were cultured in a collagen-coated silicon chamber. GCaMP6 fluorescence intensity was measured and oscillation frequency per 5 min was counted; (**D**) Trace of Ca^2+^ response in HeLa cells expressing hit GPCRs.

## Data Availability

The original contributions presented in this study are included in the article/Appendix A. Further inquiries can be directed to the corresponding authors.

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
