# Peer review of "Ligand-Independent Spontaneous Activation of Purinergic P2Y6 Receptor Under Cell Culture Soft Substrate"

_cells, 2025, doi:10.3390/cells14030216_

Round 1
Reviewer 1 Report
Comments and Suggestions for Authors
The authors are recommended to explain that why the Hela cell was selected for this experiment because cervical cancer was not mentioned in this MS.
Reviewer 2 Report
Comments and Suggestions for Authors
In this study, the authors used transfected HeLa cells to demonstrate that the spontaneous activity (i.e., in the absence of ligand binding) of several Gq protein-coupled receptors including the purinergic P2Y6 receptor depends on the stiffness of the substrate and the presence of extracellular calcium. The basal activity of the P2Y6R and other receptors was identified as induction of spontaneous Ca2+ oscillations when the cells were cultured onto silicon chambers.
The search for the molecular mechanisms of the basal activity of G protein-coupled receptors is important to understand the function of these receptors. However, I have some questions/concerns regarding this work:
Methods:
- The HeLa cells were obviously only transiently transfected. How did the authors prove successful transfection? The authors need to demonstrate the efficacy of the transfection process.
- According to the provided description, the co-transfection with GFP apparently took place with separate plasmids. This would make GFP unsuitable as a transfection control. The authors need to describe the methods in more detail and explain how they showed that the cells that exhibited Ca2+ oscillations also expressed the transfected receptor.
- The method of intracellular Ca2+ imaging with GCaMP6 is only inadequately described and not supported by references. Please describe the method in more detail.
Results and Discussion:
- Only about 30% of the cell populations exhibited Ca2+ oscillations. How can this be explained? Was this the proportion of successfully transfected cells? This needs to be clarified.
- For all experimental approaches, only three independent experiments were carried out. How many cells were evaluated per experiment (cell population)?
- The Ca2+ oscillations shown in different cells were partly synchronous and partly not. Was this possibly related to the cell density? Could it be that the proportion of cells with spontaneous Ca2+ oscillations was actually much lower and the signals were transmitted to neighboring cells via gap junctions, for example? In the uploaded supplementary AVI file, for example, this might be the case for some cell clusters.
- It has been demonstrated that the spontaneous Ca²⁺ oscillations depend on intact G-protein coupling and the presence of extracellular calcium. Which Ca²⁺ channels are involved? Inhibitors targeting diverse channels should be evaluated.
What is the reason for the absence of contribution of Ca²⁺ release from endogenous stores to the Ca²⁺ oscillations, given the requirement of intact G protein coupling?
- If the mechanism in question is related to the detection of shear stress, it should be tested using an appropriate cell culture technique. The question would be: Does the number or frequency of Ca2+ oscillations increase under shear stress?
- Could the occurrence of spontaneous Ca2+ oscillations be specific to silicone as a substrate? For example, cultivating cells in 3D hydrogels composed of native extracellular matrix is also a viable option. If the hypothesis of this work is correct, Ca2+ oscillations should also occur under such conditions.
- Finally, experiments should also be carried out with primary cells that express, for example, P2Y6 receptors to ascertain whether Ca2+ oscillations can also be observed in these cells.
Minor:
Line 51: The sentence should begin with “P1 receptors are adenosine receptors …..”
Line 237 and 325-329: Please give a reference for integrin binding to the RGD motif.
Reviewer 3 Report
Comments and Suggestions for Authors
In this manuscript Nishimura and group studied “Ligand-independent spontaneous activation of purinergic P2Y6 receptor under cell culture substrate”. Here the investigators utilized various surface matrix for cells to adapt and grown, so that the oscillations of ligand independent intracellular calcium response can be monitored. This is a very useful study and informative for studies pertaining to signal transduction pathway of G protein coupled receptor family.
(1) Authors described about that they utilized two mutants of the purinergic receptor P2Y6R, (P2Y6R AAY and P2Y6R RFD mutant). What are the expression levels and characterization of these mutants?. Elaborate the characterization of these constructs and whether expression level match the changes observed.?
(2) How the effect of laminin or Fibronectin on Gs coupled action of activation of cAMP response? How specific the extracellular matrix act on one signaling event than other cascade of signaling pathways?
Round 2
Reviewer 2 Report
Comments and Suggestions for Authors
The authors have responded appropriately to all my concerns. I recommend accepting the manuscript.